# Long-term assessment of ecosystem services at ecological restoration sites using Landsat time series

**Trinidad del Río-Mena**[1]*, **Louise Willemen**[1], **Anton Vrieling**[1], **Andy Snoeys**[2], **Andy Nelson**[1]

**1** Faculty of Geo-Information Science and Earth Observation (ITC), University of Twente, Enschede, The Netherlands, **2** Independent Consultant, The Netherlands

* t.delrio@utwente.nl, delriom.trini@gmail.com

**Data Availability Statement:** Yes - all data is available without restriction The input GIS files and the command line application to calculate the BACI contrast, together with the resulting GIS data layers

## Abstract

Reversing ecological degradation through restoration activities is a key societal challenge of the upcoming decade. However, lack of evidence on the effectiveness of restoration interventions leads to inconsistent, delayed, or poorly informed statements of success, hindering the wise allocation of resources, representing a missed opportunity to learn from previous experiences. This study contributes to a better understanding of spatial and temporal dynamics of ecosystem services at ecological restoration sites. We developed a method using Landsat satellite images combined with a Before-After-Control-Impact (BACI) design, and applied this to an arid rural landscape, the Baviaanskloof in South Africa. Since 1990, various restoration projects have been implemented to halt and reverse degradation. We applied the BACI approach at pixel-level comparing the conditions of each intervened pixel (impact) with 20 similar control pixels. By evaluating the conditions before and after the restoration intervention, we assessed the effectiveness of long-term restoration interventions distinguishing their impact from environmental temporal changes. The BACI approach was implemented with Landsat images that cover a 30-year period at a spatial resolution of 30 meter. We evaluated the impact of three interventions (revegetation, livestock exclusion, and the combination of both) on three ecosystem services; forage provision, erosion prevention, and presence of iconic vegetation. We also evaluated whether terrain characteristics could partially explain the variation in impact of interventions. The resulting maps showed spatial patterns of positive and negative effects of interventions on ecosystem services. Intervention effectiveness differed across vegetation conditions, terrain aspect, and soil parent material. Our method allows for spatially explicit quantification of the long-term restoration impact on ecosystem service supply, and for the detailed visualization of impact across an area. This pixel-level analysis is specifically suited for heterogeneous landscapes, where restoration impact not only varies between but also within restoration sites.

are available from the DANS Easy database, DOI: 10.17026/dans-zrc-hmz4.

**Funding:** The University of Twente funded the research in the form of salaries for authors Del Río-Mena, Willemen, Vrieling, and Nelson, but did not have any additional role in the study design, data collection and analysis, decision to publish, or preparation of the manuscript. Snoeys received no funding for this work, his contributions were voluntary. The specific roles of the authors are stated in the 'author contributions' section.

**Competing interests:** We confirm that there are no known conflicts of interest associated with this publication. The commercial affiliation (independent consultant) does not alter our adherence to PLOS ONE policies on sharing data and materials.

# 1. Introduction

Rural landscapes depend on and simultaneously supply several ecosystem services, nature's contribution to people [1, 2]. However, the quality of rural landscapes is deteriorating due to the expansion of croplands and grasslands into native vegetation and unsustainable agricultural practices [3]. Land degradation affects 40% of the agricultural land on earth, reducing the provision of ecosystem services and resulting in adverse environmental, social, and economic consequences [4–6]. It has been estimated that land degradation has a detrimental effect on 3.2 billion individuals and reflects an economic loss in the range of 10 percent of annual global gross product [3]. Given the increased pressure on ecosystems, restoration of degraded lands has become an important element of multiple global initiatives [7]. Several international initiatives have developed strategic targets as part of land sustainability agendas [e.g., 8–13] that are directly or indirectly linked to ecological restoration [14]. More recently, the United Nations (UN) declared 2021–2030 as the Decade on Ecological Restoration, with the aim of recognizing the need to accelerate global restoration of degraded ecosystems to mitigate negative impacts of climate change crisis protect biodiversity on the planet [15]. However, ineffective restoration efforts could inadvertently lead to a major waste of resources, continued deterioration of biodiversity and perceptions of conservation failure [16].

Restoration is defined as "any intentional activity that initiates or accelerates the recovery of an ecosystem from a degraded state" regardless of the form or intensity of degradation [17]. Restoration actions can vary from improving vegetation cover [e.g., 18, 19] to diverse land management and policy implementations for improving the quality of terrestrial [e.g., 7, 20, 21], aerial [e.g., 22], or aquatic ecosystems [e.g., 23, 24]. A successful ecological restoration should be effective, efficient and engaging through a collaboration of multiple stakeholders across sectors [25]. However, the basis of evidence to guide restoration practitioners is scarce, given the lack of long-term monitoring to determine the circumstances under which restoration efforts work [26]. This lack of impact evidence leads to incomplete, overdue or poorly informed claims of progress, hinders the effective allocation of resources and represents a lost opportunity to select the best technologies and methods based on a critical evaluation of the lessons learned [27–29].

Monitoring and evaluating restoration interventions presents several challenges, including: i) restoration projects are often implemented across large areas with limited accessibility and large spatial heterogeneity; ii) the high economic costs and capacity constraints of field monitoring methodologies hinder the long-term documentation of restoration projects, particularly to assess the effects of such interventions outside their project timespan; iii) restoration initiatives often take a long time to start generating benefits [30]; iv) simple comparison of means between impact and control sites do not account for pre-existing differences between sites; v) after a restoration effort, ecosystem services show great variation in their temporal and spatial patterns and rate of change of the trajectories towards the desired reference [31]; and vi) observed state changes may also be attributable to intra- and inter-annual climate variability, making a direct comparison of conditions before and after insufficient [32].

Remote sensing (RS) plays an important role in studying complex interactions between natural and social systems [33], such as land management. RS provides a range of data with varying spatial and temporal extents, and resolutions, facilitating monitoring and mapping of the environment [34–36]. Time series derived from satellite data can identify both rapid and longer-term changes in vegetation cover [36, 37]. Spectral information from optical satellite images allows for the quantitative estimation of biophysical vegetation characteristics, such as vegetation cover [38], aboveground biomass [e.g., 38, 39], leaf area, and leaf chlorophyll concentration [e.g., 39] among others. This remotely sensed information can be used to quantify,

map, and monitor provisioning, regulating, and (to a lesser extent) cultural ecosystem services [40]. With over 30 years of directly comparable satellite observations, freely available Landsat imagery with moderate frequency (16 days) and medium resolution (30 meter) can assess long-term dynamics of ecosystems [41, 42] and allow for temporal comparisons of restoration sites [41].

Despite widespread awareness of the potential of RS, most ecosystem service studies use static land use/land cover information rather than a more dynamic assessment of satellite records [35, 37, 41, 42]. However, land use/land cover maps are not always available [43], can lead to generalization errors as they exclude spatial variation within the same vegetation category [35, 44], and may be outdated or available only at large temporal intervals. In addition, most studies fail to take advantage of the long temporal records of available remotely sensed data, one of their great strengths in assessing ecosystem services [33, 35]. Previous studies have shown that directly linking *in situ* observations of ecosystem services to remotely sensed data improves the capturing of their spatio-temporal dynamics as compared to the often-used practice of linking the service supply directly to one land cover class [45, 46].

A method for assessing the impacts of natural or human-induced disturbances on ecosystems where the allocation of treatment and control sites cannot be randomized, is the before-after-control-impact/treatment (BACI) approach [47]. Among other applications, the BACI approach can be used to assess the impacts of long-term restoration interventions independently of natural temporal changes [48]. It compares the conditions of a restored area (impact) with the conditions of nearby unrestored (control) areas before and after the restoration intervention [49, 50]. The BACI approach was recently applied using RS images to assess land restoration interventions in semi-arid landscapes in West-Africa [51] using 20 automatically selected control sites for each impact site and multiple years for their 'before' and 'after' periods. The use of several controls invalidates claims that the findings of the BACI assessment are driven by the choice of control sites [52]. In the respective study for West Africa [33], topographic variations were not explicitly accounted for, and intervention effectiveness was assessed for the entire impact site without considering terrain variation and within-site differences in interventions' effectiveness. These can however be important because different vegetation types grow in locations with different elevation, slope, aspect, and soil parent material (geological material from which soils are formed) [53–57]. The freely accessible collection of historical Landsat imagery can mitigate the widespread lack of timely, long-term, reliable, and homogeneous ground information for monitoring restoration interventions.

This study contributes to a better understanding of the spatial and long-term distribution of ecosystem service supply for supporting the site-specific evaluation of restoration interventions by expanding the spatial scope of the BACI analyses to pixel-level. By analyzing intervention impact at 30 meter pixel-level rather than for large intervention areas, we aim to capture variation and patterns of intervention outcomes within a heterogeneous landscape. The specific aims of this research are to: (1) quantify the effect of restoration interventions on ecosystem service supply using Landsat time series data and the BACI approach at pixel-level, and (2) evaluate whether terrain characteristics affect the spatial distribution of restoration effectiveness, using an arid agricultural landscape study area in South Africa as a case study.

## 2. Materials and methods

### 2.1. Study area and interventions

The study area is composed of the subtropical arid thickets and shrublands that are in the central and eastern area of the Baviaanskloof Hartland Bawarea Conservancy, Eastern Cape in South Africa (Fig 1). The considered intervened and control areas cover about 100 km$^2$. This

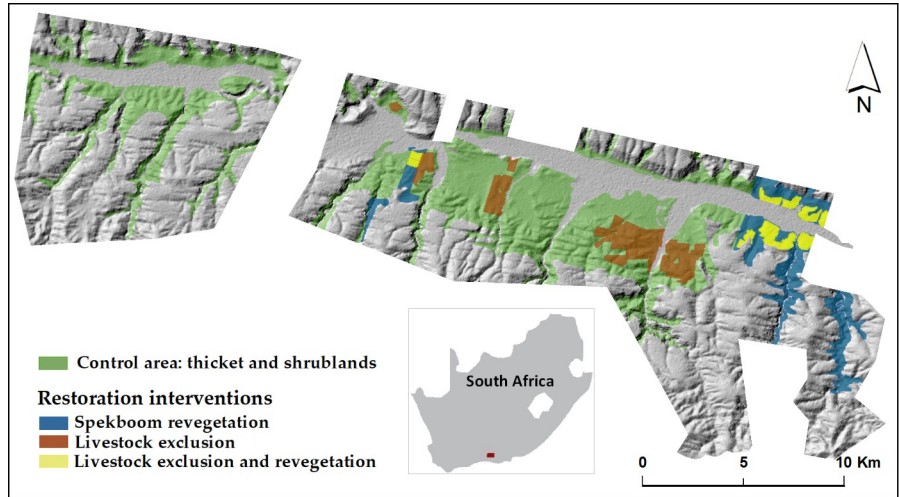

**Fig 1. Restoration intervention sites in the in the Baviaanskloof Hartland Bawarea Conservancy study area in South Africa.** Shading indicates topographic relief.

hilly region consists of a mixture of large, private farms (between 500 and 7,600 hectares in size) and rural communal land. The mean annual rainfall is 327 mm over the last 30 years, with an erratic distribution across and within years [58] (S1 and S2 Figs in S1 File). The average annual temperature in the area is 17˚C. Temperatures of up to 40˚C are frequently reported for December to February, whereas temperatures between June and August may occasionally fall below 0˚C [59].

Dense thicket vegetation in the study area is dominated by spekboom (*Portulacaria afra*), which is grazed by small livestock and wildlife [60]. Most of the thicket has been heavily degraded by unsustainable pastoralism [61, 62]. Because spekboom is a succulent species that propagates vegetatively [63], spontaneous recovery does not occur in heavily degraded sites [64, 65]. Land degradation has resulted in severe and widespread soil erosion [59]. The reduction of the natural vegetation, which is the common source of food for the extensively farmed goat and sheep in the area, has also contributed to a dramatic decline in agricultural returns in recent years [66] and degradation of the aesthetic appeal of the landscape [67]. For more than a decade, the planting of spekboom cuttings has been practiced as a practical method of restoration in the area [68–71]. Several farmers in the study area are transitioning from extensive goat and sheep farming to more sustainable farming activities such as essential oil production and agrotourism. Essential oil production is considered a more sustainable farming practice in the area as it requires limited water and fertilizer inputs and needs less land to be profitable, compared to goat farming. This transition is made in partnership with Commonland, Grounded, and Living Lands, which are three local and international non-governmental organizations. These organizations support large-scale and long-term restoration and sustainable land use initiatives.

We assessed three restoration interventions:

1. **Spekboom revegetation**: Between 2010 and 2015, about 1,100 ha were planted with spekboom to reduce degradation trends and assist the recovery of the degraded thicket vegetation [69]. The planting of spekboom truncheons was implemented through the national Department of Environmental Affairs, Natural Resource Management directorate, Expanded Public Works Program (EPWP).

2. **Livestock exclusion**: This intervention covers approximately 7,400 ha of farmland where livestock has been removed from 1990 onwards to allow for natural revegetation that could potentially prevent erosion and provide forage among other ecosystem services.

3. **Combination of livestock exclusion and revegetation:** Over time, spekboom was planted in some of the livestock exclusion areas (337 ha approximately). We considered the combination of these two ecological restoration measures as a separate intervention.

Each of the restoration interventions aimed to address local environmental challenges associated with land degradation by improving ecosystem service supply. To illustrate, for this paper we selected three ecosystem services; one provisioning, one regulating, and one cultural (Table 1).

## 2.2. General workflow

Our workflow is summarized in Fig 2. The first stage consisted of a) building ecosystem service models based on field measurements, remotely sensed spectral indices derived from Landsat 8 Operational Land Imager (OLI) and terrain variables (slope and elevation), and b) selecting the spectral index from the best-fit model as a proxy for each ecosystem service. As the terrain variables are assumed to remain constant over time, we only used the selected spectral indices to represent changes in ecosystem service supply. In the second stage, to ensure that both impact and control pixels have similar ecosystem service provision before any intervention occurred, we divided the landscape into five vegetation clusters. This clustering was based on the similarity of the changes in ecosystem service indicator under consideration. These change trajectories were derived from the ten Landsat images in the 1989–1990 period, i.e., before the intervention took place. This was achieved by applying the ISODATA unsupervised classification for the Landsat time series acquired before any intervention was implemented. In the third stage, we selected one satellite image per analyzed year based on the highest vegetation indices and lowest bare soil index separately for each ecosystem service. In the fourth stage, we estimated the intervention impact for each pixel using BACI contrast for each ecosystem service at the restoration sites. Finally, the resulting BACI contrast values were analyzed to spatially evaluate the intervention impacts, and to assess if this impact was different for vegetation clusters, terrain aspect classes, and soil parent material.

## 2.3. Model calibration and selection of spectral indices to represent ecosystem services

During the fieldwork period from May to July 2017, we estimated ecosystem services based on measurements in 32 plots of 900 $m^2$ that were distributed over the study area. We purposely selected the plots to have a large spread of values for the ecosystem service indicators, in order to better allow for extrapolation beyond the May-July period [74]. Field measurements included canopy dimensions and vegetation cover. We calculated stratified vegetation cover to

**Table 1. Selected ecosystem services and indicators used to describe their variability.**

| Ecosystem service | Ecosystem service indicator |
| --- | --- |
| Erosion prevention (regulating) | Stratified vegetation cover index (% Str.VC) [72] |
| Forage provision (provisioning) | Green biomass (kg m$^{-2}$) [73] |
| Presence iconic species (cultural) | Spekboom cover (%) [74] |

Str.VC stands for stratified vegetation cover.

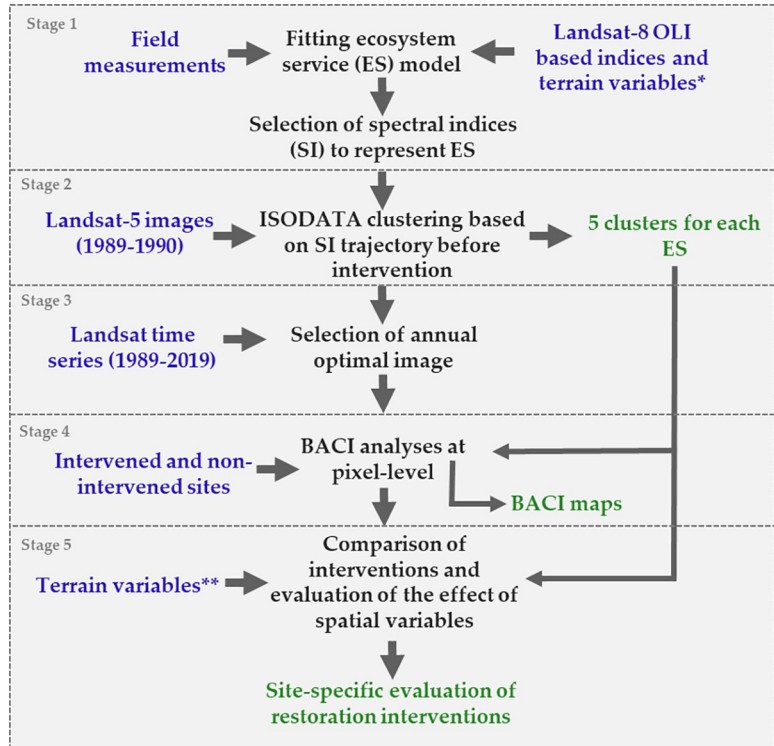

**Fig 2. Workflow diagram.** The methodological steps are referred to as stages (in grey). Blue colors indicate input data; black shows processes and analyses; green corresponds to output data. * In stage 1 we used slope and altitude. ** In stage 5 we used slope and altitude in the correlation analyses, and aspect and soil parent material for the nonparametric test.

quantify erosion prevention [72] by combining the field-measured fractional vegetation cover of different vegetation types. The presence of iconic species was estimated from field-based estimations of spekboom cover. For the forage provision we used previously developed allometric equation to estimate green biomass based on measured vegetation cover for grasses and shrubs [73].

We fitted linear and non-linear regression models using ten spectral indices (Table 2) derived from Landsat 8 OLI acquired on 14/05/2017 and terrain variables to identify a relationship between RS and field-based estimates of the ecosystem services. To avoid multi-collinearity between predictor variables, we only considered models having a variance inflation factor (VIF) lower than 5.0 [75] using the R caret package [76]. For each ecosystem service, we selected the most representative Landsat spectral index based on the best performing model according to the lowest Akaike Information Criterion (AIC) [77], using the multi-model inference (MuMIn) R package [78]. We used a five-fold cross-validation to test the models, which we repeated 100 times [79] using the 'caret' R package [76]. We used a cross-validation approach, as our limited sample size did not allow us to hold back data for independent model validation. Finally, we used another Landsat 8 OLI image matching the fieldwork period (17/07/2017) to check for consistency of the prediction of the fitted ecosystem service model at a different moment.

## 2.4. RS- GIS data description

The RS and other spatial data used in this study are summarized in Table 3 and organized according to their respective methodological stage (Fig 2). All used images from Landsat 5, 7,

**Table 2. Spectral indices with their references and corresponding equations used to explain ecosystem services measured in the field.**

| Index | Index equation |
|---|---|
| Normalized Difference Vegetation Index (NDVI) [80] | (NIR—Red) / (NIR + Red) |
| Soil Adjusted Vegetation Index (SAVI) [81] | ((NIR—Red) / (NIR + Red + L)) * (1 + L) |
| Modified Soil Adjusted Vegetation Index (MSAVI) [82] | $(2 * NIR + 1 - sqrt((2 * NIR + 1)^2 - 8 * (NIR-R))) / 2$ |
| Enhanced Vegetation Index (EVI) [83] | EVI = G* ((NIR—Red) / (NIR + C1 * Red–C2 * Blue + L)) |
| Normalized Pigment Chlorophyll Ratio Index (NPCRI) [84] | (Red—Blue) / (Red + Blue) |
| Bare Soil Index (BSI) [85] | (SWIR1 + Red)–(NIR + Blue) / (SWIR1 + Red) + (NIR + Blue) |
| *Normalized Burned Ratio (NBR) [86] | (NIR–SWIR2) / (NIR + SWIR2) |
| Normalized Burned Ratio 2 (NBR2) [86] | (SWIR1 –SWIR2) / (SWIR1 + SWIR2) |
| Normalized Difference Moisture Index (NDMI) [87] | (NIR—SWIR1) / (NIR + SWIR1) |
| Normalized Difference Water Index (NDWI) [88] | (Green—NIR) / (Green + NIR) |

NIR = Near infrared; SWIR = short wave infrared; the SWIR 1 band measures radiation in the 1.57–1.65 μm wavelength domain and SWIR 2 in 2.11–2.29 μm; G (gain factor) = 2.5; Coefficients L = 1, C1 = 6, C2 = 7.5.
*Note that NBR is sometimes named differently and is not only used for detecting burned areas, i.e., Infra-Red Index, Normalized Difference Infrared Index and Shortwave Vegetation Index [89].

and 8 were from path 172 and row 83, and accessed through Google Earth Engine (GEE). We used the Landsat Level-2 Surface Reflectance Science Product, courtesy of the U.S. Geological Survey [90], which are derived from the Landsat Collection 1 Tier 1 dataset. All selected images were cloud-free in the restoration sites (Fig 1) and small clouds were masked out in control areas. Scenes having more than 5% clouds in the control areas were excluded. For each retained image, we then extracted the relevant spectral indices. The dates of the selected images are listed on S1 and S2 Tables in S1 File. In addition to the RS data, we used elevation (meters above sea level), slope (degrees), aspect and soil parent material maps (S3–S6 Figs in S1 File).

## 2.5. ISODATA clustering, BACI analyses, and intervention evaluation

To locate areas having similar vegetation characteristics within thicket vegetation before any of the assessed interventions started, we performed an ISODATA clustering using the multi-temporal trajectory of the selected index for each ecosystem service [93]. The clustering was

**Table 3. Spatial input data for methodological stages.**

| Methodological Stage | Variables | Description | Data source |
|---|---|---|---|
| Stage 1 | 10 spectral indices | Landsat- 8 OLI from 14/05 and 17/07, 2017 | USGS [90] |
|  | Slope and elevation | 12.5 meter resolution DEM derived from ALOS PALSAR | Geophysical Institute of the University of Alaska Fairbanks [91] |
| Stage 2 | Time series of spectral indices | Extracted from Landsat 5 TM (26/02/1989 to 27/10/1990) | USGS [90] |
| Stage 3 | Spectral index values | Landsat 5, 7 and 8 images. Period 'before': 1989, 1990; 2007 to 2014 (depending on intervention). Period 'after': 2017–2019 | USGS [90] |
| Stage 4 | Restoration sites | Type and lifespan | Provided by Living Lands |
| Stage 5 | Slope, aspect, and elevation | 12.5 meter resolution DEM derived from ALOS PALSAR | Geophysical Institute of the University of Alaska Fairbanks [78] |
|  | Soil parent material | In the study area: black shale, shale, Enon conglomerate, feldspathic sandstone, quarzitic sandstone, alluvium, and terrace gravel | South African Council for Geoscience [92] |

performed in ERDAS IMAGINE using the ten available cloud free Landsat 5 Thematic Mapper (TM) images from 26/02/1989 to 27/10/1990 (S1 Table in S1 File) based on how similar the trajectory of the spectral index was between pixels. We used up to 50 iterations and a convergence limit of 1. We arbitrarily limited the prior vegetation characteristics to five vegetation clusters with the intention of distinguishing the key groups with varying temporal behavior before the interventions took place, following the procedure described in [51] (S3 Fig in S1 File).

To calculate the BACI contrast for the approximately 22,600 intervened pixels we selected Landsat images representing the greenest moment of the year. This moment is defined by calculating the average highest vegetation index -or lowest bare soil index value- for the study area (i.e., maximum MSAVI or minimum BSI value) (S2 Table in S1 File). Pixels falling within the Landsat 7 Enhanced Thematic Mapper Plus (ETM+) Scan Line Corrector (SLC) off data were excluded. We considered three years for the period before and after intervention, with exception of the interventions 'livestock exclusion' and 'livestock exclusion + revegetation' for which not enough images were available before 1989 and consequently we used only two years for the period 'before'. To focus the comparison on sites with a similar reference state, the BACI analyzes was carried out separately for each cluster. Secondly, for each of the intervened pixels, we obtained the spectral index values for each of the assessed years. We randomly selected 20 control pixels per intervened pixel [51] from the same vegetation cluster as the intervened site, avoiding pixels within the SLC off data from Landsat 7 ETM+. We also extracted the spectral index values for each intervened pixel and its respective control pixels for each year of the period before and after the intervention. We then calculated the BACI contrast based on the following formula:

$$\text{BACI contrast} = (\mu_{CA} - \mu_{CB}) - (\mu_{IA} - \mu_{IB})$$

where μ is the temporal (selected years) and spatial (20 controls) mean of the variables selected to represent the impact (in this study the selected vegetation indices); the letters C and I stand for control and impact, respectively; and the letters B and A stand for the periods 'before' and 'after', respectively. A negative contrast indicates that the variable (except the BSI index, which is a proxy for percentage of bare soil instead of vegetation cover) has increased more (or decreased less for BSI) in the impact pixel compared to the control pixels during the time period ranging from before to after the implementation of the restoration intervention (i.e., a negative BACI contrast indicates a positive intervention effect). The BACI contrast of erosion prevention (BSI) was multiplied by -1 to simplify the visualization of the map and the understanding of the ANOVA analysis. The BACI contrast is expressed in the same units of the variable of interest, i.e., the spectral index used, and consequently is unitless in our case.

Since our data did not pass the Shapiro–Wilk test for normality, we used a nonparametric Kruskal-Wallis test (Games-Howell post-hoc test at 0.05) to explore the differences between restoration interventions, vegetation clusters, terrain aspect, and soil parent material on the BACI contrast. We randomly sampled pixels for each compared group (i.e., intervention, cluster number, aspect or soil parent material), applying a minimum distance of 60 meter between points to avoid selecting neighboring pixels and ensure independent samples (S5–S7 Figs in S1 File). We selected the five soil parent materials classes with the largest area coverage in the intervened sites (Table 4). We also checked for association between the BACI contrast of each restoration intervention and ecosystem service with slope and elevation by fitting regression models, using random samples of 10% of the data.

**Table 4. Selected classes of soil parent material for BACI contrast comparison with the largest area coverage in the intervened sites.**

| Classes of soil parent material | Description |
|---|---|
| Brownish quartzitic sandstone | Brownish weathering quartzitic sandstone, fine to coarse grained, shale |
| Feldspathic sandstone | Impure feldspathic sandstone, subordinate shale |
| Shale | Black shale, subordinate siltstone |
| Enon conglomerate | Conglomerate: sandstone, siltstone, clay |
| Whitish quartzitic sandstone | Whitish-weathering quartz sandstone; feldspathic near top, subordinate shale; cross-bedded; med-coarse grained |

# 3. Results

## 3.1. Selection of spectral indices

The RS-based models that best describe ecosystem services as measured in the field are presented in Table 5. The best model for erosion prevention is a second-degree polynomial fit with the BSI index. Forage provision was also fitted to second-degree polynomials using the NBR. The presence of iconic species is the only ecosystem service described with a linear regression model that uses two predicting variables (MSAVI and elevation), where elevation contributed by 22% to the model (expressed as partial $R^2$). The $R^2$ in Table 5 represents the mean $R^2$ obtained from the repeated cross-validation. We used another image matching the fieldwork period to check if the models were consistent for different dates. The models based on spectral information for 17/07/2017 resulted in lower $R^2$ as compared to when spectral information for the validation data14/05/2017 was used for erosion prevention and forage provision. The $R^2$ of presence of iconic species increased by 2% for the validation date.

## 3.2. RS-based BACI analyses

The maps showing the BACI contrast of the assessed ecosystem services of approximately 22600 intervened pixels at 30 meter resolution are presented in Fig 3. All the assessed ecosystem services show similar general patterns of positive and negative effects of interventions. However, when zooming in, at pixel level their BACI contrasts show differences. For correct interpretation of the results for erosion prevention, it is important to note that the BSI index shows bare soil cover and was multiplied by -1 to allow the same BACI contrast interpretation as for the other assessed ecosystem services. Therefore, negative BACI contrast values for BSI also indicate a positive effect on erosion control of the intervention. In this section, we use the term 'better' to indicate lower BACI contrasts. Even though in most areas the BACI contrast

**Table 5. Selected ecosystem service models based on indices derived from calibration between field measurements with Landsat 8 (14/05/2017) data and terrain variables.**

| Ecosystem service | Function | $R^2$ | Standardized RMSE[*] | n | $R^2$ validation date (17/07/2017) |
|---|---|---|---|---|---|
| Erosion prevention | StrVC = 56.36(BSI$-^2$−36.66(BSI) + 6.06 | 0.85 | 0.19 | 32 | 0.71 |
| Forage provision | GB = 47.62(NBR)$^2$ + 55.55(NBR) + 8.71 | 0.71 | 1.19 | 32 | 0.67 |
| Presence of iconic species | SbC = 12.24(MSAVI) + 0.01(Elevation) - 10.49 | 0.53 | 0.62 | 20 | 0.54 |

StrVC = Stratified vegetation cover, GB = Green biomass, SbC = Spekboom cover.

[*] Std. RMSE = RMSE / (Ymax- Ymin), where Ymax and Ymin correspond to the maximum and minimum measured values of the ecosystem service indicator.

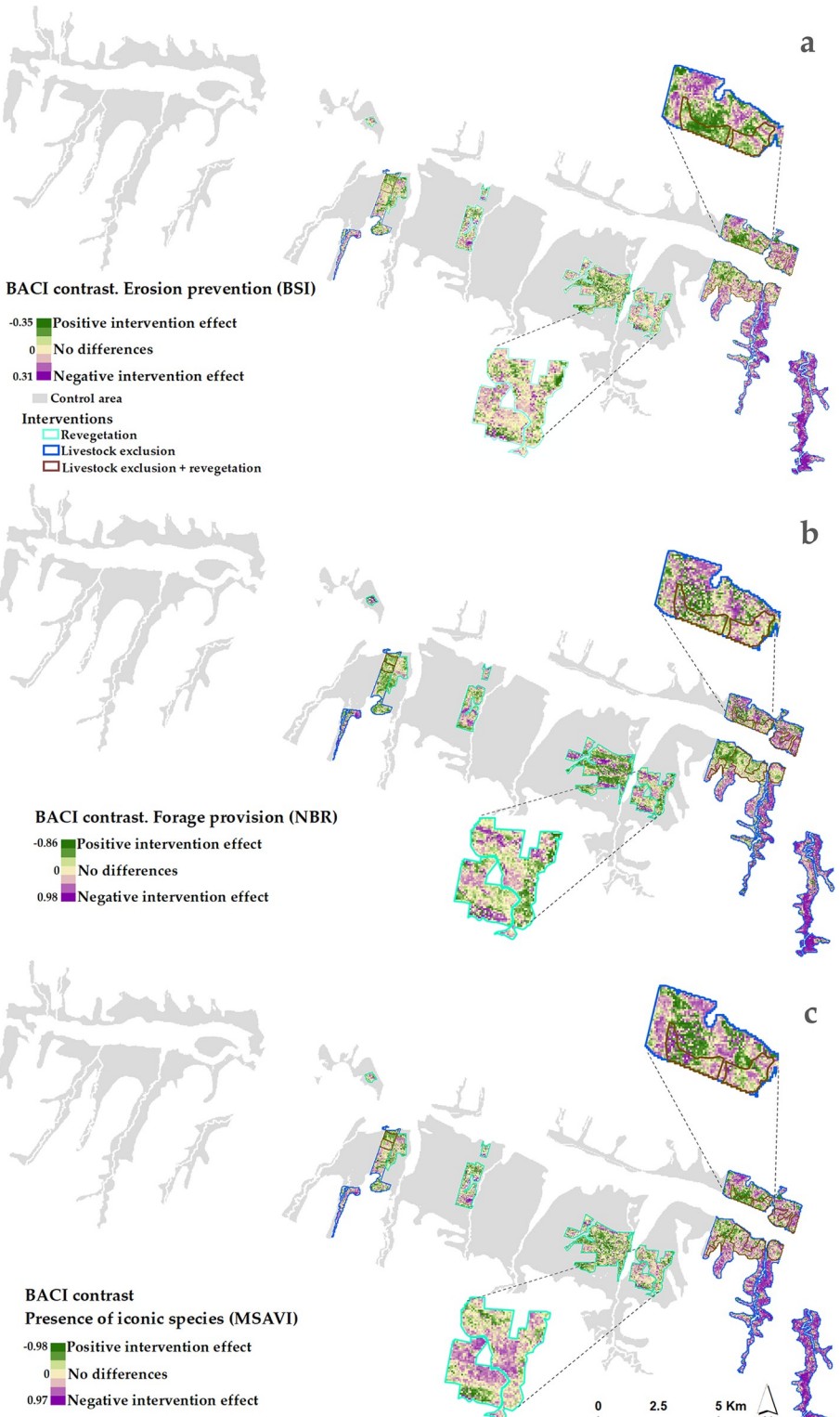

**Fig 3.** BACI contrast at pixel-level to assess the effect of each restoration intervention in the study area on: a) erosion prevention (based in the BSI index); b) forage provision (based on NBR index); and c) presence of iconic species (based on MSAVI index). The thicket and shrublands area used to for selecting control sites is indicated in grey.

presents similar patterns for the three ecosystem services, the magnified areas in Fig 3 also show areas with clear differences in BACI contrast for the different ecosystem services. In addition, the maps show high spatial variability of restoration interventions impact within restoration sites. Within a single intervention, ecosystem service supply both increased and decreased.

## 3.3. Comparison of BACI contrast between interventions, vegetation clusters and terrain variables

In Table 6 the differences in BACI contrast between restoration interventions for the different ecosystem services are presented, as a measure of impact of these interventions. The interquartile range (IQR, the difference between the 25th and 75th percentile) is presented as indication of variability of the BACI contrast within these intervention sites. The lower this value, the more homogenous the intervention effect. The Hodges-Lehmann estimator of the median was used as the non-parametric indicator for each group [94], estimating a "pseudo–median" for non-symmetric populations, which is closely related to the population median [95].

The post-hoc test indicated that 'revegetation' sites are not significantly different from 'livestock exclusion + revegetation' sites for erosion prevention. Both areas did not present evident impact on erosion prevention, while the BACI contrast for 'livestock exclusion' sites even showed a negative effect on this ecosystem service (Table 6). For forage provision, areas with livestock exclusion performed significantly better (negative BACI contrast) as compared with revegetated areas and livestock exclusion combined with revegetation. Whereas for the presence of iconic species, 'revegetation' sites did not show intervention impact while the other two interventions presented positive BACI contrasts, meaning deterioration after intervention.

We also evaluated the relation between the state of the vegetation before interventions took place (as indicated by their cluster) and the BACI contrast. Vegetation clusters 1 and 2 comprised areas with relatively more vegetation, whereas vegetation clusters 4 and 5 contain more bare soil. Improvements in BACI contrasts (negative) were found in more vegetated areas (clusters 1 and 2) in 'revegetation' sites for erosion prevention, although only cluster 2 performed significantly better. Cluster 2 also showed better BACI contrast for forage provision. In contrast, clusters 4 and 5 showed no clear difference after the intervention (Table 7). Cluster 1 presented an improvement in presence of iconic species in all the intervened areas. We excluded cluster 5 to compare 'revegetation' sites for the presence of iconic species because the area was too small resulting in insufficient independent pixels to run the comparison. Among all the evaluated

**Table 6. Comparison of BACI contrast between interventions at pixel-level using the Kruskal-Wallis test (Games-Howell post-hoc).**

| Landsat index | Represented ecosystem service | Intervention | Hodges-Lehmann estimator | IQR | Post-hoc* |
|---|---|---|---|---|---|
| BSI | Erosion prevention | Revegetation | 0.00 | 0.06 | a |
| | | Livestock exclusion | 0.04 | 0.10 | b |
| | | Livestock exclusion + revegetation | 0.00 | 0.06 | a |
| NBR | Forage provision | Revegetation | 0.01 | 0.10 | a |
| | | Livestock exclusion | -0.04 | 0.15 | b |
| | | Livestock exclusion + revegetation | 0.00 | 0.07 | a |
| MSAVI | Presence of iconic species | Revegetation | 0.00 | 0.07 | a |
| | | Livestock exclusion | 0.02 | 0.11 | b |
| | | Livestock exclusion + revegetation | 0.02 | 0.07 | b |

IQR = interquartile range.

*different letters indicate significant differences of Games-Howell post-hoc at p-value < 0.05

**Table 7. Comparison of BACI contrast between vegetation clusters indicating the state of the vegetation for each intervention and ecosystem service using the Kruskal-Wallis test (Games-Howell post-hoc).**

| Intervention | Landsat index | Represented ecosystem service | Cluster | Hodges-Lehmann estimator | IQR | Post hoc[*] |
|---|---|---|---|---|---|---|
| Revegetation | BSI | Erosion prevention | 1 | -0.02 | 0.06 | ab |
| | | | 2 | -0.02 | 0.06 | a |
| | | | 3 | 0.01 | 0.05 | c |
| | | | 4 | 0.00 | 0.06 | b |
| | | | 5 | 0.00 | 0.04 | b |
| | NBR | Forage provision | 1 | 0.03 | 0.59 | abcd |
| | | | 2 | -0.08 | 0.21 | a |
| | | | 3 | -0.01 | 0.15 | b |
| | | | 4 | 0.02 | 0.12 | c |
| | | | 5 | 0.00 | 0.08 | d |
| | MSAVI | Presence of iconic species | 1 | -0.02 | 0.14 | ab |
| | | | 2 | 0.00 | 0.03 | ab |
| | | | 3 | -0.01 | 0.07 | a |
| | | | 4 | 0.01 | 0.07 | b |
| Livestock exclusion | BSI | Erosion prevention | 1 | 0.06 | 0.10 | a |
| | | | 2 | 0.06 | 0.09 | b |
| | | | 3 | -0.03 | 0.07 | c |
| | | | 4 | 0.01 | 0.08 | d |
| | | | 5 | 0.01 | 0.04 | d |
| | NBR | Forage provision | 1 | 0.03 | 0.25 | ab |
| | | | 2 | -0.05 | 0.15 | c |
| | | | 3 | -0.05 | 0.15 | a |
| | | | 4 | -0.01 | 0.12 | b |
| | | | 5 | 0.03 | 0.06 | d |
| | MSAVI | Presence of iconic species | 1 | -0.04 | 0.22 | a |
| | | | 2 | 0.04 | 0.08 | b |
| | | | 3 | 0.01 | 0.10 | d |
| | | | 4 | 0.00 | 0.07 | d |
| | | | 5 | 0.01 | 0.05 | c |
| Livestock exclusion + revegetation | BSI | Erosion prevention | 1 | -0.01 | 0.11 | a |
| | | | 2 | 0.02 | 0.07 | b |
| | | | 3 | -0.02 | 0.05 | c |
| | | | 4 | 0.00 | 0.06 | a |
| | | | 5 | 0.00 | 0.04 | a |
| | NBR | Forage provision | 1 | 0.04 | 0.23 | abcd |
| | | | 2 | -0.04 | 0.13 | a |
| | | | 3 | -0.02 | 0.08 | b |
| | | | 4 | 0.00 | 0.06 | c |
| | | | 5 | 0.02 | 0.06 | d |
| | MSAVI | Presence of iconic species | 1 | -0.05 | 0.21 | a |
| | | | 2 | 0.05 | 0.08 | b |
| | | | 3 | 0.03 | 0.06 | c |
| | | | 4 | 0.03 | 0.07 | c |
| | | | 5 | 0.00 | 0.05 | d |

NSD = No significant differences between groups. IQR = interquartile range. Cluster numbers are ordered from initial high to low vegetation cover (e.g., cluster

1 = denser vegetation).

[*]different letters indicate significant differences of Games-Howell post-hoc at p-value < 0.05

restoration interventions sites, the variability of forage provision in cluster 1 (expressed by the IQR) was considerably higher than for the other clusters. This cluster represents 4.2% of the total intervened pixels. For the 'livestock exclusion' intervention, the BACI contrast indicated a deterioration of the erosion prevention service (as shown by a positive Hodges-Lehmann estimator value) in clusters representing denser vegetation before the intervention began (i.e., clusters 1–2). Forage provision improved in clusters 2 and 4 at 'livestock exclusion' sites, while presence of iconic species showed an increase only on cluster 2. From Table 7, we can observe that especially areas that had little vegetation before the restoration intervention (Cluster 4–5) showed no change or a slight decrease in erosion prevention and forage provision, while several of other vegetation clusters improved these ecosystem service supply.

Regarding the assessed terrain variables to explain the intervention impact, we found that the terrain aspect did not consistently explain the BACI contrast values for the three ecosystem services (details are presented in S3 Table in S1 File). The comparison of the BACI contrasts for the classes of soil parent material showed that the only improvements were found for forage provision in feldspathic sandstone and whitish quartzitic sandstone (S4 Table in S1 File). For the three ecosystem services, the BACI contrasts for sites with brownish quartzitic sandstone differed most from sites with other soil parental types. This parent material specifically showed worse BACI contrasts (positive) for all ecosystem services (S4 Table in S1 File). This soil parent material represents an average of 51% of all pixels. Regression models between BACI contrast with elevation and slope did not show any clear association, with $R^2$ lower than 0.05 for all ecosystem services and interventions (S5 Table in S1 File).

## 4. Discussion

### 4.1. Selection of spectral indices

Our RS-based models suggest that spectral indices extracted from Landsat images can help to quantify the supply of the studied ecosystem services in the region. The indices that best captured the ecosystem services in the study area are based on the blue, red, near-infrared (NIR), and short-wave infrared (SWIR 1) wavelengths. Since we used the stratified vegetation cover as erosion prevention indicator, the BSI could effectively reflect the lack of this cover. Besides the BSI, the NBR index also presented a good fit with erosion prevention. Although NBR is usually used to detect burned areas, the index (NIR − SWIR)/(NIR + SWIR) has been referred to with different terminology [89] and as such has been also used to estimate wet and dry biomass [96], forest harvest [97], gross primary production [98] and vegetation water content [99].

Even though the presence of iconic species is linked to the percentage of spekboom cover, the model did not capture the presence of one single species as precisely (53% of the spekboom cover variation) as models that include the overall presence of green vegetation (StrVC and green biomass). In agreement with [46], the integration of elevation improved the capacity of the RS-model to capture the presence of iconic species, explaining 22% of the variance not captured by the Landsat 8 index. The model suggests that the presence of spekboom increases with elevation within the intervened area (between 380 and 680 meter above sea level). Among other reasons, elevations of 390 meter and lower are on the frost-prone valley floor and could impair the growth of spekboom [53].

The estimation of RS-based ecosystem services using field measurements provide higher accuracy and lower site-specific errors than estimations based on individual land cover types [45, 100]. However, temporal extrapolation of such models with RS indices requires validation with data across time to decrease the uncertainty in the evaluation of restoration interventions. In this study, we tested our fitted models with other images acquired during the same

fieldwork period. However, we could not validate the stability of the relationship between the spectral images and ecosystem services in the field for the full 30-year period.

## 4.2. RS-based BACI analyses

The BACI approach allows for relative comparisons of spatial and temporal differences that can be used to extract the unbiased restoration impact [52, 101]. However, correct understanding of the underlying calculation process is needed to accurately interpret results. First, for each restoration intervention, BACI was applied using a different number of years for the pre-intervention period (S2 Table in S1 File). The choice of this reference value will affect the BACI contrast, given that the spectral indices values vary between years [51], and consequently the result of the BACI contrast (i.e., the impact of restoration interventions) are also affected by the specific conditions of that selected year. For example, droughts or rainy years will affect vegetation cover and therefore the values of spectral indices. Assuming that climatic conditions are rather homogeneous in the neighborhood of the restored sites, this problem was partially solved by comparing the conditions of the restoration area before and after the intervention with those of similar areas nearby. Also, taking more than one year into account for the period 'before' and 'after' of the BACI calculations helps to compensate for index inter-annual variation, because it ensures that coincidental temporal variations do not restrict the identification of the effects [50]. Secondly, each pixel is assigned a p-value that shows the significance of the BACI contrast using 20 control sites (e.g., S9 Fig in S1 File). We compared 20 control pixels for each intervened pixel. These p-values could change, or non-significant BACI contrasts can become significant when changing the number of controls to, for example, 100 pixels. A larger number of control sites would be more time and computational consuming, and our explorative analysis with 100 control sites resulted in similar interventions impacts (S6 Table in S1 File).

The pixel-level implementation of BACI using RS data assists in better spatially explicit evaluations of restoration interventions. Our method is particularly efficient for collecting historical data and evaluating large, remote, and heterogeneous areas where data collection is difficult and resource consuming. Our study area is located in a dry area with relatively little cloud cover. The availability of RS images will decrease in areas with frequent cloudy days, such as the in humid tropics. Depending on the availability of satellite images, the 'before' and 'after' reference periods could be changed or extended, allowing, for example, for the exclusion of abnormally dry years, or adding another after period to evaluate the differences at different intervals of times from the start of the intervention.

Although the levels of the BACI contrast are small when expressed as absolute numbers (e.g., forage provision ranging from 0.04 to -0.11 in Table 6, translating to -9 to 11 kg per m$^2$) they can represent high relative change in ecosystem supply. The relative BACI contrast (as presented in S8 Fig in S1 File) highlights the magnitude of the contrast in relation to the analyzed pixel's spectral index value before the intervention took place. The percentages in the map of S8 Fig in S1 File are particularly high when the baseline spectral value was close to zero.

## 4.3. Comparison of BACI contrast between interventions, vegetation clusters and terrain variables

Our presented pixel-level restoration evaluation method is specifically useful to evaluate heterogeneous landscapes, where restoration impact not only varies between restoration sites but also within. The inclusion of several control sites, a multi-year period 'before' in the BACI analyses, and its comparison with the current state allowed capturing the spatial and temporal effects of interventions otherwise invisible. We learned for example that livestock exclusion

showed on average a positive impact compared to control sites and to 30 years ago. The maps in Section 4.2 visualize how this impact is spatially distributed within the intervened sites in order to guide future management of the area. The inclusion of vegetation clusters representing the status of vegetation before the intervention began, allowed to visualize if areas that originally had more vegetation responded better or worse to the intervention than areas having little vegetation. Livestock exclusion or this intervention combined with revegetation showed a positive effect on the presence of native iconic species in densely vegetated areas (cluster 1). The better intervention results in more vegetated areas for the presence of the iconic species under livestock exclusion could be explained by considering that spekboom only propagates vegetatively [63]. Therefore, spontaneous recovery will not occur in heavily degraded areas [65]. For erosion prevention and forage provision, negative or no intervention effect were also found for these degraded areas. These findings are in line with other assessments that also found that more severely degraded areas have lower restoration successes [102]. In our study we could not compare the BACI contrast to a quantified target value the restorations interventions aimed that. While we observed changes ecosystem services supply, without a target value we cannot make statements of intervention success. Regardless, more research would be needed to confirm this behavior and understand if there are other reasons of why originally vegetated areas responded more poorly to interventions than areas having little vegetation.

The inclusion of aspect and soil parent material allowed capturing differences and gaining insights for the interpretation of intervention impacts and potentially guiding intervention actions. Depending on the context of the interventions and specific monitoring objectives, the BACI contrast comparison could be improved by including other variables in the analyses. For example, by aspect and soil parent material of each intervention; year of when the revegetation started (from 2010 to 2015), or the inclusion of other fine scale data (e.g. high-resolution historical climate data). Variables correlated with land use intensity and past land use trends, may also influence spatial heterogeneity of the restoration effect [103].

## 5. Conclusions

The new evaluation method presented in this paper allows for mapping and quantifying the long-term impact of restoration interventions on ecosystem services at pixel-level. Our approach helps to differentiate the intervention effect from other environmental factors and is especially useful for monitoring large, remote, and heterogeneous landscapes. The resulting maps visualize spatial patterns of intervention effects within larger intervention sites and provide insights on their temporal changes. As such, these maps help to learn from restoration experience and mitigate ineffective future restoration efforts. Moreover, our approach allows us to identify which spatially variable factors may explain the success or failure of an intervention. By measuring restoration impact on different ecosystem services, we increase our understanding of social benefits and trade-offs of restoration choices. The presented approach can be extended to a broader range of restoration interventions and ecosystem services in different contexts across different landscapes, as long as spectral indices and spatial indicators can be identified to represent these ecosystem services. Our RS-based approach is particularly suitable to estimate changes in ecosystem services directly linked with the presence of vegetation and less of ecosystem services that are strongly linked to a specific species or human perception, for which additional spatial data would be needed. Our understanding of the well RS-based ecosystem service models perform over multi-year periods can be improved by having long-term research sites to validate relations between field observations and spectral information over time.

## Supporting information

**S1 File.**
(DOCX)

**S1 Raw images.**
(PDF)

## Acknowledgments

We are grateful to members of Living Lands in the Baviaanskloof Hartland Bawarea Conservancy, South Africa for support in providing network and logistic facilitation, providing crucial background data, assistance for fieldwork facilitation (especially from Elwin Malgas, Luyanda Luthuli, Melloson Allen, and the interns Jurian Schepers and Amanda Alfonso-Herrera), providing working facilities and the friendly and enabling environment. We extend our appreciation to Michele Meroni, for clarifying methods to carry out the analyses of intervention restorations. Finally, we would like to thank all the farmers involved for their cooperation, friendliness, accessibility, and knowledge sharing during fieldwork.

## Author Contributions

**Conceptualization:** Trinidad del Río-Mena, Louise Willemen.

**Data curation:** Trinidad del Río-Mena.

**Formal analysis:** Trinidad del Río-Mena.

**Investigation:** Trinidad del Río-Mena.

**Methodology:** Louise Willemen, Anton Vrieling, Andy Nelson.

**Software:** Andy Snoeys.

**Supervision:** Louise Willemen, Anton Vrieling, Andy Nelson.

**Visualization:** Trinidad del Río-Mena.

**Writing – original draft:** Trinidad del Río-Mena.

**Writing – review & editing:** Louise Willemen, Anton Vrieling, Andy Snoeys, Andy Nelson.

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
