## [Decision Letter · Decision Letter 0]

17 Mar 2021

PONE-D-20-35781

Long-term assessment of ecosystem services at ecological restoration sites using Landsat time series

PLOS ONE

Dear Dr. del Río,

Thank you for submitting your manuscript to PLOS ONE. After careful consideration, we feel that it has merit but does not fully meet PLOS ONE’s publication criteria as it currently stands. Therefore, we invite you to submit a revised version of the manuscript that addresses the points raised during the review process. The comments are appended below. In addition to this, please also address the comments given in the attached file.

We look forward to receiving your revised manuscript.

Kind regards,

Shijo Joseph, Ph.D.

Academic Editor

PLOS ONE

Journal Requirements:

2) Thank you for stating the following in the Financial Disclosure section:

[The authors received no specific funding for this work.].   

We note that one or more of the authors are employed by a commercial company: Independent Consultant

i. Please provide an amended Funding Statement declaring this commercial affiliation, as well as a statement regarding the Role of Funders in your study. If the funding organization did not play a role in the study design, data collection and analysis, decision to publish, or preparation of the manuscript and only provided financial support in the form of authors' salaries and/or research materials, please review your statements relating to the author contributions, and ensure you have specifically and accurately indicated the role(s) that these authors had in your study. You can update author roles in the Author Contributions section of the online submission form.

ii. Please also provide an updated Competing Interests Statement declaring this commercial affiliation along with any other relevant declarations relating to employment, consultancy, patents, products in development, or marketed products, etc. 

3) We note that you have stated that you will provide repository information for your data at acceptance. Should your manuscript be accepted for publication, we will hold it until you provide the relevant accession numbers or DOIs necessary to access your data. If you wish to make changes to your Data Availability statement, please describe these changes in your cover letter and we will update your Data Availability statement to reflect the information you provide.

4) Please upload a copy of Figures4 & 5, to which you refer in your text on line 317. If the figure is no longer to be included as part of the submission please remove all reference to it within the text.

Reviewers' comments:

Reviewer's Responses to Questions

**Comments to the Author**

1. Is the manuscript technically sound, and do the data support the conclusions?

Reviewer #1: Yes

2. Has the statistical analysis been performed appropriately and rigorously? 

Reviewer #1: Yes

3. Have the authors made all data underlying the findings in their manuscript fully available?

Reviewer #1: Yes

4. Is the manuscript presented in an intelligible fashion and written in standard English?

Reviewer #1: Yes

5. Review Comments to the Author

Reviewer #1: This paper describes a topical study showcasing how publicly available remote sensing data can be combined with field work data to assess the impact of ecosystem restoration operations.

I was a bit unclear about stage 2 (figure 2). What is the objective of clustering based on the trajectory? Also, what do you mean by “trajectory”? Can you put in 2-3 sentences explaining this step.

Please clarify this point: In section 2.3, you have described a model that links ~May Landsat image reflectances to field data. If phenology is an important factor in your study area, then the fact that the indices you have chosen best represent the ES variables would be applicable in May. But then you use images “representing the greenest moment of the year”. At this moment, the “may-month-based indices” may not be the best.

If possible, do a more rigorous error analysis, so that spatial estimates of error/uncertainty is made.

I have several minor comments in the attached PDF: “PONE-D-20-35781_reviewerComments.pdf”. Otherwise, the methodology seems okay and the results are fine. Good work!

6. PLOS authors have the option to publish the peer review history of their article (what does this mean?). If published, this will include your full peer review and any attached files.

Reviewer #1: No

---

## [Author Response · Author response to Decision Letter 0]

23 Apr 2021

Long-term assessment of ecosystem services at ecological restoration sites using Landsat time series

Response to the Editor

>> We appreciate the consideration of our paper for revision. Thank you for the helpful feedback and corrections. 

>> In addition to the revisions regarding the editor and the reviewer feedback, we found and corrected a mistake in the calculation and interpretation of the BACI contrast in the results and discussion. These corrections resulted in a modification of the sign of the BACI contrast (Legend in Figure 3), although the location of the positive and negative effect of the interventions did not change. We also corrected the estimates of the RS model for presence of iconic species (Table 5). This change did not affect any of the results as we only used the vegetation index for the BACI analysis. 

Please ensure that your manuscript meets PLOS ONE's style requirements

>> In this version, we adjusted the text format according to the formatting guidelines.

Regarding Financial Disclosure section

Please provide an amended Funding Statement declaring this commercial affiliation, as well as a statement regarding the Role of Funders in your study.

>> We now included the Funding Statement as indicated in the cover letter.

Please provide un updated Competing Interests Statement declaring this commercial affiliation along with any other relevant declarations relating to employment, consultancy, patents, products in development, or marketed products, etc.

>> We now updated the Competing Interests Statement in the cover letter as requested.

We note that you have stated that you will provide repository information for your data at acceptance. Should your manuscript be accepted for publication, we will hold it until you provide the relevant accession numbers or DOIs necessary to access your data. If you wish to make changes to your Data Availability statement, please describe these changes in your cover letter and we will update your Data Availability statement to reflect the information you provide.

>> Thank you for this reminder. We are working on uploading the repository information in Data Archiving and Networked Services (DANS) and will provide the DOI access information to be included in the accepted manuscript.

Please upload a copy of Figures4 & 5, to which you refer in your text on line 317. If the figure is no longer to be included as part of the submission please remove all reference to it within the text.

>> We corrected this error in the reference of Figure 3.

Response to Reviewer 1

This paper describes a topical study showcasing how publicly available remote sensing data can be combined with field work data to assess the impact of ecosystem restoration operations.

>>We would like to thank the reviewer for the constructive and comprehensive feedback that helped us to improve the quality of this manuscript. We address and respond to the specific comments below.

I was a bit unclear about stage 2 (figure 2). What is the objective of clustering based on the trajectory? Also, what do you mean by “trajectory”? Can you put in 2-3 sentences explaining this step.

>>We edited the text in order to clarify the ISODATA classification and the terminology used. 

Lines 213-219 it now reads: “In the second stage, to ensure that both impact and control pixels have similar ecosystem service provision before any intervention occurred, we divided the landscape into five vegetation clusters. This clustering was based on the similarity of the changes in ecosystem service indicator under consideration. These change trajectories were derived from the ten Landsat images in the 1989-1990 period, i.e. before the intervention took place.”

Please clarify this point: In section 2.3, you have described a model that links ~May Landsat image reflectances to field data. If phenology is an important factor in your study area, then the fact that the indices you have chosen best represent the ES variables would be applicable in May. But then you use images “representing the greenest moment of the year”. At this moment, the “may-month-based indices” may not be the best.

>>Indeed, we unfortunately lack temporal field data for different seasons to estimate variations of ES estimations using the RS models. However, to calibrate the models we used plots having different levels of ES supply (i.e. different vegetation cover, green biomass and spekboom cover percentage). This is described in our previous work referenced in line 238 (Del Río-Mena et al., 2020). For more clarification the first sentence of section 2.3 now reads: “During the fieldwork period from May to July 2017, we estimated ecosystem services based on measurements in 32 plots of 900 m2 that were distributed over the study area. We purposely selected the plots to have a large spread of values for the ecosystem service indicators, in order to better allow for extrapolation beyond the May-July period [77].”

If possible, do a more rigorous error analysis, so that spatial estimates of error/uncertainty is made.

>>In the section 4.3 of the Discussion we describe the implications of our data choices and possible errors. We now better clarify the uncertainty resulting from the selection of the reference years (line 612). In this section, we now also better indicate the implications of the control site selection for the BACI outcomes and significance (line 624). To explore to the sensitivity to the number of control sites, we run a number of analyses for 100 controls instead of 20, these did not result in notable differences (line 628) (Table S6). 

SPECIFIC COMMENTS PROVIDED AS COMMENT BOXES IN THE MANUSCRIPT

1. Introduction

Lines 85: Repeated words

>>We deleted the repeated words “and vi)” 

2.1. Study area and interventions

Figure 1: Order of restoration interventions 

>>The restoration interventions in the legend are now organized according to the order in the text.

Figure 1: Add rectangle to the main map so that it corresponds to the small red rectangle in the inset (bottom left).

>>We added the rectangle as indicated. Although the red rectangle in the inset shows an indication of the location of the study area and it does not show the exact size, we now adjusted the dimensions to make them closer real size.

Table 1: From the table is not clear the provisioning, regulating, and cultural ecosystem service

>>We now added the type of ecosystem service in brackets in the first column of Table 1.

2.3. Model calibration and selection of spectral indices to represent ecosystem services

Table 2: Add references to the indices 

>>We now added the source references to each ecosystem service. 

2.4. RS- GIS data description

Table 3: Use full name “soil parent material”

>>We now changed “parent “material” for “soil parent material” for the whole manuscript. 

2.5. ISODATA clustering, BACI analyses, and intervention evaluation

Line 264-265: This detail is not needed: Just say that you selected control from the same vegetation cluster.

>>We edited the sentence. It now reads, “We randomly selected 20 control pixels per intervened pixel (Meroni et al., 2017) from the same vegetation cluster as the intervened site, avoiding pixels within the SLC off data from Landsat 7 ETM+.”

3.1. Selection of spectral indices

Table 5: Define “standardized RMSE, provide the %RMSE and give formula in the text. 

>>To be able to compare the residual variance between ecosystem services models (different units and scales), we standardized the RMSE results using the formula Std. RMSE = RMSE/(Ymax- Ymin).

We now added this explanation below Table 5 as “* Std. RMSE = RMSE / (Ymax- Ymin), where Ymax and Ymin are the maximum and minimum measured values of the ecosystem service indicator.”

References 

Del Río-Mena, T., Willemen, L., Vrieling, A., Nelson, A., 2020. Remote sensing for mapping ecosystem services to support evaluation of ecological restoration interventions in an arid landscape. Ecol. Indic. 113, 106182. https://doi.org/10.1016/j.ecolind.2020.106182

Meroni, M., Schucknecht, A., Fasbender, D., Rembold, F., Fava, F., Mauclaire, M., Goffner, D., Di Lucchio, L.M., Leonardi, U., 2017. Remote sensing monitoring of land restoration interventions in semi-arid environments with a before–after control-impact statistical design. Int. J. Appl. Earth Obs. Geoinf. 59, 42–52. https://doi.org/10.1016/j.jag.2017.02.016

---

## [Editor Report · Decision Letter 1]

28 Apr 2021

Long-term assessment of ecosystem services at ecological restoration sites using Landsat time series

PONE-D-20-35781R1

Dear Dr. del Río,

We’re pleased to inform you that your manuscript has been judged scientifically suitable for publication and will be formally accepted for publication once it meets all outstanding technical requirements.

Kind regards,

Shijo Joseph, Ph.D.

Academic Editor

PLOS ONE

---

## [Editor Report · Acceptance letter]

3 Jun 2021

PONE-D-20-35781R1 

Long-term assessment of ecosystem services at ecological restoration sites using Landsat time series  

Dear Dr. del Río-Mena:

I'm pleased to inform you that your manuscript has been deemed suitable for publication in PLOS ONE. Congratulations! Your manuscript is now with our production department. 

Kind regards, 

on behalf of

Dr Shijo Joseph 

Academic Editor

PLOS ONE